# Nanotechnology Strategies in Plant Genetic Engineering: Intelligent Delivery and Precision Editing

**DOI:** 10.3390/plants14233625

**Published:** 2025-11-28

**Authors:** Chun-Mei Lai, Xiao-Shan Xiao, Li-Wei Liu, Xin-Da Lin, Dan-Lin Dou, Han-Yang Cai, Zhi-Feng Mei, Fan Yang, Yan Cheng, Yuan Qin

**Affiliations:** 1 College of Life Sciences, Fujian Agriculture and Forestry University, Fuzhou 350002, China; 2College of Plant Protection, Fujian Agriculture and Forestry University, Fuzhou 350002, China; 3Fujian Provincial Key Laboratory of Haixia Applied Plant Systems Biology, Fujian Agriculture and Forestry University, Fuzhou 350002, China

**Keywords:** nanotechnology, gene delivery, CRISPR-Cas genome editing, sustainable agriculture

## Abstract

Plant genetic engineering is crucial for enhancing crop yield, quality, and resilience to both abiotic and biotic stresses, thereby promoting sustainable agriculture. Agrobacterium-mediated, biolistic bombardment, electroporation, and poly (ethylene glycol) (PEG)-mediated genetic transformation systems are widely applied in plant genetic engineering. However, these systems have limitations, including species dependency, destruction of plant tissues, low transformation efficiency, and high cost. Recently, gene-delivery methods based on nanotechnology have been developed for plant genetic transformation. This nanostrategy demonstrates remarkable transformation efficiency, excellent biocompatibility, effective protection of exogenous nucleic acids, and the potential for plant regeneration. However, the application of nanomaterial-mediated gene-delivery systems in plants is still in its early stages and faces numerous challenges for widespread adoption. Herein, the conventional genetic transformation techniques utilized in plants are succinctly examined. Subsequently, the advancements in nanomaterial-based gene-delivery systems are reviewed. The applications of CRISPR-Cas-mediated genome editing and its integration with plant nanotechnology are also examined. The innovations, methods, and practical applications of nanomaterial-mediated genetic transformation summarized herein are expected to facilitate the progress of plant genetic engineering in modern agriculture.

## 1. Introduction

Genetic transformation stands as a pivotal domain within functional genomics research and molecular breeding in the realm of plant science [1]. The quest for safe, efficient, and innovative genetic transformation methodologies has consistently been a focal point in genetic engineering, molecular biology, and genetic breeding [2]. Traditional plant genetic transformation approaches encompass Agrobacterium tumefaciens infection [3], particle bombardment [4], and virus infection [5]. These methods have thus far seen extensive utilization in model plants such as Arabidopsis thaliana [6] and tobacco [7], as well as in annual or biennial herbs [8] and horticultural plants [9]. Due to the rapid advancement of nanoscience and nanotechnology in the late 20th century, nanomaterials have gained widespread application in nanobiology and gene therapy. This is attributed to their small size, large surface area, biocompatibility, biodegradability, low toxicity, and low immunogenicity [10]. Recently, a nanomaterial-mediated gene-delivery system has been developed. This system achieves high transformation efficiency in plant cells without the need for external physical or chemical means, demonstrating excellent potential for applications in plant genetic engineering [11].

Despite significant advancements in plant genetic engineering, it continues to lag behind the progress made in animal genetic engineering [12]. The plant cell wall acts as a significant barrier, impeding direct access to the cell membrane and thus hindering many manipulations in plant genetic engineering [13]. It comprises cellulose, hemicellulose, and pectin, which play roles in maintaining cell shape, boosting mechanical strength, and mediating plant cell immunity [14]. The plant cell wall permits only biomolecules with a diameter smaller than 20 nm to pass through, which impedes the entry of exogenous biomolecules into plants [15]. In addition to the cell wall, plant cell membranes, nuclear membranes, and organelle membranes also act as barriers for foreign biomolecules to cross. Consequently, genetic transformation in many plants remains challenging and is often unsuccessful.

Based on the unique properties of nanomaterials, this review summarizes the types of gene carriers used in plant genetic transformation, their methods of combining with foreign genes, and the advantages over traditional transgenic techniques (Figure 1). It also discusses the challenges and future prospects of nanomaterial-mediated gene delivery to inspire the development of optimized and innovative plant genetic transformation technologies. A comparison of the strengths and limitations of different genetic transformation methods is provided in Table 1.

## 2. Bottleneck Landscape of Conventional Gene-Delivery Systems

### 2.1. Agrobacterium Transformation

*Agrobacterium tumefaciens* (crown gall bacterium) and *A. rhizogenes* are common Gram-negative soil bacteria; *A. tumefaciens* can infect more than 140 plant species and induce crown gall disease. Through the tumor-inducing plasmid (Ti plasmid), a T-DNA segment is transferred into plant cells under the mediation of virulence proteins (Vir proteins), the T-DNA traverses the cell wall and plasma membrane [16] and ultimately integrates randomly into the plant nuclear genome. Exogenous genes are delivered via vacuum infiltration, injection, or floral dip. Dicotyledonous plants (embryos with two cotyledons), as natural hosts, have been successfully transformed: in 1977, Chilton et al. first demonstrated stable integration of virulent plasmid DNA into higher-plant cells [17]; in 1985, the leaf-disk method developed by Horsch expanded the range of recipient species [18] and was applied to tobacco [19], sorghum [20], potato [21], tomato [22], blueberry [23], and soybean [24]. Transformation efficiency is regulated by chemical signal molecules that activate Vir genes (such as acetosyringone synthesized by dicot cell walls) [25]. Subsequently, breakthroughs in monocot transformation were achieved in the 1990s: Hernalsteens first achieved genomic transformation in asparagus [26]; Grimsly introduced a maize streak virus gene into maize and induced systemic infection [27]; Hiei identified rice cultivar, culture conditions, bacterial strain virulence, and promoter choice as core factors for transformation efficiency [28], and the technique was then extended to rice [29], barley [30], wheat [31], and maize [32]. Three requirements must be met for successful transformation: adsorption of *Agrobacterium* to the plant cell wall, induction of Vir-gene expression by plant signal molecules, and the presence of competent host cells [33]. Currently, more than 85% of transgenic crops are constructed using this method [34]. Advantages include operational simplicity, low cost, and the ability to deliver protein–DNA complexes for nuclear targeting [35]. However, there are notable limitations: monocots are difficult to transform because they naturally lack the biosynthetic pathway for the signal molecules, and random integration readily causes unintended agronomic traits. In recent work, the Kan Wang team combined a modified ternary Vir helper plasmid (carrying an extra virA) with a thymidine auxotrophic strain, increasing *Agrobacterium* transformation efficiency in maize B104 from 25.6% to 33.3% and reducing the risk of over-proliferation, thereby providing a promotable high-efficiency *Agrobacterium* delivery scheme [36].

### 2.2. Gene Gun Delivery: Tissue Damage and Risk of Multicopy Integration

In 1987, the Sanford team pioneered gene gun technology [37]. The gene gun uses high-velocity tungsten microprojectiles to penetrate the cell wall and plasma membrane, introducing DNA or RNA into intact cells and triggering transient expression; in principle, it can bypass the host-range limitation of *Agrobacterium* and the regeneration bottleneck of protoplasts, but stable transformation of small, regenerable cells still remains to be demonstrated. In 1988, Klein delivered exogenous DNA into intact maize cells [38], validating monocot applicability; the next year, the Mendel team used high-speed microprojectile bombardment to deliver exogenous DNA into barley suspension cells and detected high-level transient expression of the GUS and NPT II reporter genes within three days [39]. In 1990, Boynton restored algal photosynthetic function via chloroplast targeting [40], opening a new field of organelle engineering; in 1992, Vasil bombarded wheat embryos to obtain fertile lines (>1% transformation rate) [41], advancing applications in staple crops; in 1995, Caimi leveraged the capacity to deliver large fragments to introduce the bacterial SacB gene into maize [42], reshaping the fructose metabolic pathway. In 1997, Bidney found that pre-bombardment wounding increased *Agrobacterium* efficiency by two orders of magnitude [43], a mechanism that inspired Mesa’s technique of *Agrobacterium*-coated gold microprojectiles (1999) [44], which doubled GUS expression in strawberry. The technology was also extended to protoplasts [45], callus [46], and pollen [47]. Owing to species independence, the diversity of recipient materials, operational simplicity, high plastid (such as chloroplast) transformation efficiency, and the ability to deliver large DNA fragments up to approximately 150 kb, particle bombardment has been widely used in genetic engineering [48]. Nevertheless, its limitations are notable: on the one hand, it depends on costly instruments and consumables (gene guns and gold particles, etc.). On the other hand, large DNA tends to break during acceleration. Intracellular DNA damage is mainly repaired through two pathways—HDR (applicable to plastids and the nucleus) and NHEJ (restricted to the nucleus); meanwhile, interactions among homologous sequences can induce transcriptional/post-transcriptional silencing at the DNA–DNA, DNA–RNA, and RNA–RNA levels. Exogenous DNA also often integrates randomly into the nuclear genome in clustered arrays with rearrangements, resulting in multicopy transgenes. In response to this complexity, researchers continue to elucidate mechanisms and explore improvements [49]. In recent years, the Kan Wang team added a 3D-printed flow-guided barrel (FGB) to the front end of the Bio-Rad PDS-1000/He gene gun to optimize particle/airflow and target uniformity, increasing transient transfection in onion epidermis by 22-fold, Cas9-RNP editing by 4.5-fold (Figure 1A), and viral infection in maize seedlings by 17-fold; in immature embryos of maize B104, the stable transformation rate rose from 0.33% to 11.33%, and the overall process was shortened from 165 days to 72 days (Figure 1B). To explore its potential in wheat, we adapted a protocol to deliver DNA vectors into SAMs, aiming to generate transgenic or genome-edited wheat plants. The results showed that the average genome-editing efficiency in the T0 generation was 4.0% under the condition of three bombardments with the conventional gene gun (Condition 1), while no positive plants were generated with a single bombardment using the conventional gene gun. In contrast, Conditions 3 and 4, which employed the FGB, achieved genome-editing efficiencies of 8.7% and 9.3%, respectively, representing a twofold or greater improvement in the T0 generation (Figure 1C) [50], achieving simultaneous gains in efficiency and quality enabled by engineering optimization.

### 2.3. Electroporation/PEG Method: The Protoplast Regeneration Barrier Electroporation

Electroporation transiently increases membrane permeability with short, intense electric pulses, allowing macromolecules such as plasmid DNA (pDNA) to enter the cytosol (Figure 2A) [11]. The earliest in vitro electroporation of protoplasts was established in 1985; subsequently, by optimizing voltage, pulse parameters, and medium composition, success has been achieved in multiple mono- and dicot protoplasts [51]. The standard procedure for electroporation can be condensed into a four-step process [52]. Initially, cells are co-incubated with plasmid DNA (pDNA). Subsequently, an electric field is applied to alter membrane potential distribution. This is followed by the formation of pores in the cell membrane to facilitate pDNA entry. Finally, upon field removal, the pores close, the membrane is restored, and the expression of the introduced molecules commences.

Although the cell wall imposes size and charge barriers (approximately 5–20 nm), electroporation can still achieve delivery in intact cells. For example, Cre recombinase has been introduced into *Arabidopsis thaliana* cells, further demonstrating the feasibility of this strategy [53]. Mechanistically, Weaver et al. proposed that the electric field first induces a membrane “dimple” and creates hydrophobic pores, which then partially convert to hydrophilic pores (Figure 2B) [52]; however, these conclusions are mainly integrated from different experiments and still lack direct in situ observational evidence.

Compared with biolistics and *Agrobacterium*-mediated delivery, electroporation offers rapid procedures, lower cost, and considerable efficiency, and it is applicable to single cells and cell aggregates. Corresponding limitations include a relatively narrow recipient range and difficulty with thick-walled cells; in addition, strong fields may damage naked nucleic acids, and electrochemical reactions at the electrodes may generate end products that are toxic to protoplasts [54].

### 2.4. PEG-Mediated Delivery

Unlike the physical method of electroporation, PEG is a synthetic, biocompatible hydrophilic polyether that is widely used in pharmaceuticals [55]. In plants, it was first used to promote protoplast fusion and was later adopted to facilitate the uptake of DNA by protoplasts [56]. The typical procedure is to place protoplasts together with exogenous DNA in a PEG system and use divalent cations such as Ca^2+^ to trigger the direct transmembrane entry of DNA [57]; PEG transiently increases plasma-membrane permeability and enhances adsorption of DNA at the membrane surface, thereby allowing naked DNA to enter the cytosol [58]. However, despite high efficiency at the protoplast level, protoplast regeneration is difficult in most species, which limits its use for obtaining stable transgenic plants. In 2023, the joint team from the Yunnan Tobacco Genetic Engineering Research Center and BGI-Shenzhen achieved a practical breakthrough along the PEG-Ca^2+^ route. Using PEG-Ca^2+^ protoplast transformation to deliver CRISPR/Cas9 plasmids and donor DNA, they achieved a seamless 1819 bp replacement of the N′ gene in tobacco and obtained T0 plants resistant to TMV-U1 that lacked foreign-fragment integration. Using PEG-Ca^2+^ transformation, a seamless 1819 bp replacement in tobacco generated TMV-U1-resistant T0 plants without foreign-fragment integration, providing a direct line to stable traits (Figure 3) [59].

## 3. Nanocarrier Design: From Material Classification to Smart Responsiveness

Compared with traditional plant genetic transformation methods (such as *Agrobacterium* transformation and gene gun delivery), nanoparticle-based strategies exhibit four core advantages: low cytotoxicity that avoids membrane damage caused by high-voltage electric fields or microprojectile impact; operational convenience in that no cell-wall enzymatic pretreatment is required; species generality; and the ability to co-deliver diverse biomolecules that supports the simultaneous loading of nucleic acids, quantum dot imaging probes [60], and active regulatory molecules. Studies have shown that nanoparticle internalization efficiency (e.g., intracellular enrichment of gold nanoclusters [61]) correlates strongly with target gene expression; therefore, achieving efficient delivery requires meeting a key size constraint at least one dimension < 20 nm (for instance, carbon nanomaterials with diameters of 1–3 nm [62], nanowires with lateral dimensions < 15 nm [63]). In addition, small size confers unique benefits: nanoparticles can pass through the chloroplast envelope (<10 nm) and the nuclear pore complex, enabling precise subcellular targeting (chloroplasts, mitochondria, and nucleus, etc.). To fully realize the potential of the technology, it is urgently needed to develop new nanomaterials that combine small-size penetrability (<20 nm), smart responsiveness (pH-/enzyme-triggered release), and multistage targeting capability to improve genetic transformation efficiency.

### 3.1. Carbon-Based Nanoplatforms

#### 3.1.1. CDs

CDs are zero-dimensional carbon nanomaterials typically smaller than 10 nm, with tunable fluorescence and good water dispersibility, and a relatively clear structure–property framework has been established [64]. They were first reported in 2004 as fluorescent carbon fragments and purified by electrophoresis, laying a materials foundation for later applications [65]. With advances in surface passivation and functionalization strategies, CDs have shown broad applicability in biology, energy, and sensing, together with engineering-friendly interfacial chemistry [66]. In energy and environmental uses, CDs can act as photocatalytic platforms for efficient visible-light degradation of aqueous pollutants, reflecting good charge–carrier separation and synergistic active sites [67]. In addition, debate over whether graphene quantum dots truly exhibit upconversion luminescence has driven clarification of their photophysical mechanisms and standardized measurements [68]. Turning to plant systems, CDs can enhance light harvesting and photosynthesis and upregulate photosystem-related genes such as PsbP/PsiK, thereby improving crop growth under certain conditions (Figure 4A) [69]. For gene delivery, PEI-functionalized CDs can load plasmid DNA by electrostatic coupling and enable expression of exogenous proteins in multiple crops, offering a lightweight chemical carrier for plant molecular breeding (Figure 4B) [70]. Meanwhile, foliar spraying of siRNA-loaded CDs can achieve efficient gene silencing without windowing or wall removal, showing the feasibility of field-friendly nucleic acid delivery (Figure 4C) [71].

#### 3.1.2. CNTs

As high-aspect-ratio carbon nanomaterials, CNTs including single-walled carbon nanotubes (SWNTs) and multiwalled carbon nanotubes (MWNTs), which are widely used at biological interfaces; in mammalian systems, SWNTs have been applied in image-guided photothermal therapy, pointing to programmable optical and thermal properties and engineering potential [72]. In parallel, studies on the in vivo migration and fate of multiwalled CNTs provide essential evidence for biocompatibility and safety boundaries, laying a basis for cross-species risk assessment [73]. On the methods side, DNA and other π-affine molecules can stabilize CNT dispersions and tune their interfaces through noncovalent interactions, offering a practical route for entry into biological systems [74]. On the application side, work on CNT-based biosensors has systematically summarized device mechanisms and recent performance advances, showing strong designability and portability [75]. In addition, composites of cationic polymers such as poly (allylamine) with CNTs can increase nucleic-acid loading and interfacial stability, thereby providing materials support for subsequent gene delivery [76].

Further, in plant systems, SWNTs have been directly observed to cross the cell wall and enter intact cells, demonstrating the physical penetration and co-delivery potential of high-aspect-ratio carriers [77]; meanwhile, protoplast models reveal that uptake and subcellular localization of MWCNTs are not simple passive diffusion but are shaped by active pathways and intracellular fate [78]. For functional delivery, PEI-CNTs have enabled transient expression of plasmid DNA in multiple mature leaves without genomic integration, validating a route for “non-integrative expression” (Figure 5A) [79]; chitosan-complexed SWNTs achieve chloroplast-selective delivery and trigger target-gene expression, introducing a design concept for organelle targeting (Figure 5B) [80]; at the same time, siRNA-loaded SWNTs achieve efficient gene silencing in intact leaves, further extending non-transgenic nucleic-acid intervention [81].

Mechanistically, the LEEP model indicates that particle size and surface charge dominate membrane traversal and residence, providing a clear design basis for rational optimization of carrier parameters [82]. Methodologically, operational notes and process-oriented summaries for whole-plant materials offer direct references for implementation and condition optimization [83]. More broadly, recent reviews have mapped the routes, advantages, and limitations of nanomaterials in plant genetic engineering, helping position CNTs within the overall technical landscape [84]. In addition, macromolecules such as proteins can traverse the cell wall and membrane into plant cells with the help of nanocarriers, and early in vivo experiments demonstrated the feasibility of protein delivery [85]. Finally, frontier reviews comparing different CRISPR/Cas cargos and delivery routes emphasize designs that improve efficiency while lowering integration risk [86].

#### 3.1.3. Graphene Derivatives and Other Carbon Nanomaterials

As important carbon materials beyond CNTs, the nano–bio interface between graphene/GO and nucleic acids has been systematically elucidated: graphene/GO can achieve efficient adsorption, reversible release, and fluorescence modulation of nucleic acids via π-π and electrostatic interactions, laying the foundation for programmable carriers and sensing platforms [87]. Following this interfacial chemistry, GO–aptamer complexes can perform in situ molecular-probe detection inside living cells, exhibiting an “adsorption quenching-hybridization desorption” working mode and good compatibility with the cellular environment [88]. Meanwhile, nanodiamonds, by virtue of surfaces amenable to functionalization and good biocompatibility, can serve as multimodal drug/oligonucleotide carriers to achieve integrated functions such as selective delivery, imaging, and enhanced therapy [89]. In plant scenarios, nanodiamonds can act as projectiles and be passively delivered into plant tissues via the gene gun, suggesting that “non-CNT” carbon materials also have physical routes to enter plant cells, although related applications still require further expansion and validation [90].

### 3.2. Inorganic, Non-Carbon Carriers

#### 3.2.1. MSNs: Pore-Channel Encapsulation and Controlled Release

Silicon is the second most abundant element in the Earth’s crust. Although it is not classified as an essential plant nutrient, its nanoscale forms such as silica nanoparticles show notable potential for enhancing plant stress resistance; prior reviews have systematically summarized mechanisms and application progress in mitigating biotic/abiotic stresses [91]. Regarding entry routes, the systematic review by the burkhardt team pointed out that stomatal infiltration is co-regulated by particle size, leaf-surface wettability, and surfactants, so the “leaf surface-stomatal” path is one of the important channels for nanoparticles to enter leaves [92].

At the level of cellular uptake, Jambhrunkar et al. showed that positive charge/amine functionalization together with suitable surface wettability can markedly increase the endocytic efficiency of MSNs, indicating that a “surface-chemistry-driven” design strategy should be prioritized over simply emphasizing morphology [93]. Consistent with this, 20 nm, amine-functionalized fluorescent MSNs were taken up by wheat, lupin, and *Arabidopsis thaliana*, without obvious acute toxicity, thereby validating at the whole-plant level the feasibility of the “size-surface chemistry” combination [94].

Within the “pore-channel encapsulation and controlled release” paradigm, Torney et al. established the idea of “end-cap leakage prevention-stimuli-triggered release”. They first achieved plasmid expression in protoplasts, and then, via gold-nanoparticle capping and stimuli-induced uncapping, obtained expression in leaves and callus (for expression only, without claiming stable integration) [95]. To overcome tissue barriers and improve in vivo entry, Martin-Ortigosa et al. gold-plated MSNs to increase density and, together with bio-ballistic parameter optimization coordinated with 0.6 μm gold microcarriers, significantly enhanced entry efficiency and tissue distribution, while also indicating the cost and surface-deposition limits of the strategy [96]. Further, to reduce reliance on external forces and highlight functional nucleic-acid delivery, Cai et al. used a spraying strategy to deliver siRNA with MSNs in mature leaves, achieving long-term and multigene silencing and demonstrating, under whole-plant conditions, an integrated route of “no mechanical force-functional nucleic acids-persistent phenotype”. Foliar spraying of MSN-siRNA realized long-term multigene silencing in whole plants, illustrating a field-friendly operating mode (Figure 6A) [97].

#### 3.2.2. Innovative Applications of LDH in Plant Gene Delivery

LDH are positively charged, two-dimensional layered inorganic materials, first proposed by Choy et al. as a prototype of “non-viral vectors,” laying a materials basis for biological delivery [99]. In addition, LDH and their derivatives show excellent activity in electrochemical water oxidation (OER) and have become one of the efficient catalyst systems [100]. In catalysis, LDH, leveraging interlayer anion exchangeability and tunable metal sites, can function as solid bases/acids, supports, or precursors to regulate active centers and reaction pathways [101]. In energy-storage devices, LDH improve specific capacitance and ion/electron transport through composition control and the design of two-dimensional sheets/porous structures [102]. In analytical/environmental adsorption, LDH achieve selective enrichment and preconcentration via anion exchange and surface sites [103]. At the cellular and whole-plant levels, Bao et al. used positively charged LDH-lactate exfoliated nanosheets (thickness from 0.5 to 2 nm; lateral sized from 30 to 60 nm) to build electrically neutral complexes with fluorescent dyes and 60-mer ssDNA, and found rapid cytosolic entry within minutes to 15 min in BY-2 suspension cells and *Arabidopsis* root cells; effects were seen at 25 μg/mL with short incubation. Endocytosis inhibitors and low-temperature treatment did not block internalization, and nuclear localization of ssDNA-FITC was observed, providing direct evidence for “wall crossing-cell entry-nuclear delivery” [104].

Under intact-leaf conditions, Yong et al. achieved rapid internalization in mature leaves of *Nicotiana benthamiana* with about 40 nm LDH nanoparticles, and confirmed particle movement along the apoplast and vascular system; more importantly, LDH markedly enhanced nucleic-acid entry so that exogenous siRNA reduced the target transgene mRNA by >70% within 1 day, and chloroplast uptake was observed as an organelle-level distribution feature, indicating that the parameterized framework of “particle size-formulation-intra-tissue transport” holds in vivo [98] (Figure 6B). Within the foliar-spray paradigm of “pore-channel encapsulation and controlled release,” the BioClay (dsRNA-LDH) system proposed by Mitter et al. showed wash-off resistance with dsRNA detectable on leaf surfaces for ≥30 days, and in tested virus models, a single spray provided at least a 20-day protection window, highlighting long-acting retention and functional readouts of LDH-mediated nucleic acids [105].

#### 3.2.3. AuNCs: Photothermal Responsiveness and Enhanced Gene Silencing

Christou et al. first used an electronic gene gun to introduce gold microparticles coated with exogenous DNA into soybean seeds, establishing an early paradigm for the involvement of gold materials in plant gene introduction, but whether gold nanomaterials can autonomously cross the intact cell wall and complete delivery has remained unsettled [106]. Along the “self-delivery” route, Zhang et al. constructed 1–2 nm PEI-AuNCs, formed stable complexes with siRNA, and entered mature leaf tissues by leaf infiltration, showing good tissue compatibility and controllable delivery [61] (Figure 6C). Time-series imaging and quantification indicated continuous internalization over 20 min–24 h with a peak at 1 h, thus yielding strong readouts at both ends of “entry speed-duration” [61]. Functionally, AuNCs delivery produced significant downregulation at both mRNA and protein levels in two models, mGFP5 and ROQ1, and exhibited RNase protection and stability advantages in tissue contexts, demonstrating not only tissue entry but also maintenance of cargo bioactivity [61]. These results indicate that a formulation strategy of “ultrasmall clusters plus cationic shells” can serve as an effective chemical self-delivery route, and that key parameters (cluster size, PEI molecular weight) can be co-optimized with dosing mode to balance entry kinetics and silencing strength [61]. Zhang et al. conducted a comparative study on the impact of nanoparticle shape and size on their uptake by mature leaves. They analyzed spherical gold nanoparticles of 5, 10, 15, and 20 nm in diameter, as well as gold nanorods measuring 13 × 68 nm with an aspect ratio of approximately 5.2. The study revealed that the gold nanorods were internalized by the leaves, whereas the 10 nm spherical nanoparticles were not [107]. Surprisingly, despite no observed internalization, the 10 nm spheres showed stronger gene silencing under the same conditions, leading to the key proposition that “particle internalization is not a prerequisite for achieving silencing,” which provides direct evidence that material entry behavior and functional readout can be decoupled [107]. Mechanistically, the specific reason for this difference remains to be clarified, but the results expand the design space for gold nanomaterials in plants from “must enter cells” to “entry and effect can be optimized separately,” providing clear handles for parameterized design across morphology (aspect ratio), surface chemistry, and dosing modality [107].

### 3.3. Organic–Biological Hybrid Systems

#### 3.3.1. Liposomes: Membrane-Fusion Delivery

As vesicular systems composed of a phospholipid bilayer enclosing an aqueous core and capable of fusing with the plasma membrane, liposomes were used early in plant research to efficiently deliver TMV RNA into tobacco protoplasts, establishing a feasibility basis for nucleic-acid delivery [108]. At the whole-plant level, Karny et al. applied HSPC liposomes by foliar spraying and showed that nanoparticles penetrated into leaves and were internalized, with a particle penetration rate of about 33% (free molecules < 1%); exogenous Fe/Mg alleviated acute deficiencies in tomato and outperformed conventional fertilizers, indicating the potential of foliar dosing in crop systems (Figure 7A) [109]. In early applications, liposomes were more commonly used in protoplast systems, often validating expression of viral RNA, with the earliest functional evidence provided by Fukunaga et al. [110]. Mechanistically, Nagata systematically examined different types of liposomes (such as PS/Chol anionic, neutral PC and cationic PC/stearylamine) and their interactions with plant plasma membranes, directly observing endocytosis under PS/Chol conditions; this route differs from membrane fusion, and its molecular mechanism remains to be elucidated [111]. Methodologically, Lurquin summarized preparation routes including the reverse-phase evaporation method (REV) and reported co-delivery of TrosV RNA and pLGV23neo DNA in negatively charged liposomes, providing an operational paradigm for “dual cargo in one vesicle” [112]. Notably, in the presence of PEG and Ca^2+^, liposomes can efficiently fuse with protoplast membranes and yield stably expressing clones, showing marked gains in efficiency and reproducibility [113].

Accordingly, liposome–protoplast interactions can be summarized into three models: (1) membrane fusion (PEG/PVA + Ca^2+^ as common fusogens); (2) chemical endocytosis; and (3) formation of transient pores upon contact with the plasma membrane, leading to release of the internal contents [111]. At the ultrastructural level, Fukunaga et al. further observed intact liposomes carrying portions of plasma-membrane material within the cytosol, providing morphological support for endocytosis/membrane-sheet engulfment [114]. In non-protoplast contexts, Rosenberg et al. delivered CAT and TYLCV-related genes into tobacco/tomato callus using negatively charged large unilamellar vesicles and obtained expression readouts, demonstrating operability at the tissue level [115]. From a design-and-application viewpoint, Gad et al. systematically summarized the advantages of liposomes as plant gene-delivery carriers (such as membrane fusion-enhanced entry, protection of nucleic acids from nuclease degradation, capacity for targeting, and applicability across cell types) and pointed to potential uses in cell types such as pollen, providing a methodological framework for expanding target tissues [116].

#### 3.3.2. Chitosan-Based Nanocomposite Systems: Enhanced Cell-Wall Penetration

Modified chitosan has been widely used in biomedicine, agriculture, and food sciences; its cationic backbone forms stable complexes with nucleic acids and serves as a general carrier platform for diverse dosing and delivery scenarios [117]. At the mammalian-cell level, chitosan–pEGFP composite nanoparticles achieved stable transfection and fluorescent-protein expression in primary chondrocytes, indicating suitability for hard-to-transfect primary systems [118]. In epithelial cell lines, quaternized chitosan oligomers, as new carriers, markedly increased gene-transfection efficiency, showing a feasible path in which positive-charge tuning and hydrophilicity/sustained release act together to optimize carrier performance [119]. In vivo, direct intra-articular injection of chitosan-mediated plasmids into rabbit knee joints yielded local expression, demonstrating stability and biocompatibility in a tissue microenvironment [120]. For plant applications, a chitosan–salicylic acid nanocomposite raised anthocyanin accumulation and stress-tolerance indices in crops such as grape, indicating agronomic potential for “ligand synergy-penetration enhancement” [121]. In addition, chitosan microspheres used as urea slow-release carriers extended the nutrient-release window and smoothed peak inputs under field conditions, highlighting engineering-ready attributes for crop management (Figure 7B) [122].

It should be emphasized that current evidence still lacks confirmation that chitosan alone can achieve functional gene delivery into plant cells; however, a chitosan–SWNT system has achieved chloroplast-selective delivery with in vivo expression, suggesting that the combination of chitosan surface functionalization with high-aspect-ratio nanostructures can serve as an effective route in plant settings [80]. From a design standpoint, molecular weight and degree of deacetylation (DA), N/P ratio, surface modifications such as quaternization/PEGylation, and sustained-release formulations act together to determine condensation, cellular uptake, and expression readouts, providing a framework of tunable levers from materials to biology [123].

#### 3.3.3. Peptide Carriers: Endosomal Escape and Nuclear Targeting

Peptide carriers, as non-viral gene delivery platforms, rely on polycationic backbones to form complexes with nucleic acids, thereby improving loading and initial anti-degradation capacity and offering a feasible alternative to plant transformation routes that do not depend on *Agrobacterium* or the gene gun [124]. In mammalian cells, the transmembrane internalization mechanisms of cell-penetrating peptides (CPPs) and sequence-structure rules have been systematically summarized, providing inspiration for sequence design in plant systems [125]. Notably in plants, early studies first demonstrated that exogenous proteins can enter cells by traversing the cell wall and lipid bilayer via CPPs [126], and subsequent work achieved plasmid DNA delivery and expression under intact-tissue conditions, indicating functional output of peptide-mediated non-viral transfection [127]. Centered on coupled design of “entry-release,” stimuli-responsive peptides can trigger nucleic-acid dissociation from carriers in the mildly acidic endosomal environment and achieve more robust transfection at both the protoplast and tissue levels [128].

At the organelle level, peptide–DNA complexes have achieved selective delivery and integration of exogenous DNA into mitochondrial and chloroplast genomes, paving the way for organelle genetic improvement [129]. By combining clustered CPPs with chloroplast-targeting peptides, precise delivery to multiple plastid types and spatially controlled distribution within tissues can be achieved, showing programmable potential for three-level targeting across “tissue-organelle-locus” [130]. For nuclear delivery, integrating stimuli-responsive release promotes effective intracellular release of DNA and supports nuclear expression readouts, providing a path for nuclear entry of genome-editing components (Figure 7C) [128].

At the same time, the common endocytosis–endosome–vacuole pathway in plants limits effective release of complexes; although histidine-rich or otherwise protonatable modules can promote endosomal escape in some animal systems, plant contexts still require dedicated optimization targeting the cell wall/endosome/vacuole chain, and stimuli-responsive peptides offer a feasible strategy to improve escape and expression in plant DNA delivery [128]. Systematic screens show pronounced sequence dependence in CPP uptake across materials including *Arabidopsis*, tobacco, tomato, poplar, and rice callus, with frequent observations of vacuolar accumulation and limited escape, indicating that merely increasing entry rate is insufficient to ensure effective expression [131].

At the materials-engineering level, dual-peptide systems that decouple and recombine high-efficiency uptake with intracellular transport/release modules can raise transfection efficiency in callus and improve cytosolic transport and release performance, thereby amplifying net effects (Figure 7D) [132]. To reinforce wall traversal and transport at the tissue level, the programmable geometry and rigidity of DNA nanostructures provide structural variables for “physical barrier crossing + biological targeting,” serving as design references for peptide systems [133]; in three-dimensional tissue models, open-frame structures outperform solid structures in tissue penetration, indicating that coupling among morphology, mechanics, and tissue passability can be used to enhance deep-delivery performance of composite systems [134]. Peptide carriers furnish a programmable delivery route for plant cells that does not rely on *Agrobacterium*/gene gun and covers the closed loop of “loading-wall crossing-release-localization.” Notably, efficiency bottlenecks lie mainly in endosomal/vacuolar escape and organelle targeting and can be improved through stimuli-responsive strategies and dual/multipeptide co-design.

**Figure 7 plants-14-03625-f007:**
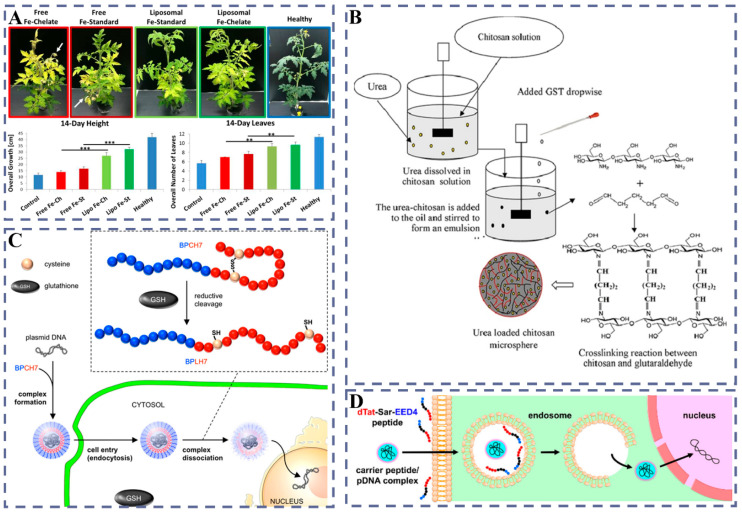
(**A**) Leaf-applied Fe liposomal supplements effectively restored plant vitality and reduced necrosis. Reprinted with permission from ref. [109], Copyright 2018, Springer Nature. (**B**) Fabrication of chitosan microspheres for controlled release of urea fertilizer. Reprinted with permission from ref. [122], Copyright 2021, Elsevier. (**C**) Illustration of the glutathione-reducible peptide (BPCH7) and the proposed mechanism for intracellular delivery and subsequent pDNA release. Reprinted with permission from ref. [128], American Chemical Society, Copyright 2018. (**D**) Dual-peptide gene-delivery system for efficient plant callus cell transfection. Reprinted with permission from ref. [132], Copy 2020, American Chemical Society. ** *p* < 0.01; *** *p* < 0.001.

#### 3.3.4. DNA Self-Assembled Structures: Programmable Nucleic-Acid Carriers

DNA self-assembly, including DNA origami, “site-wise assembles” nucleic acids and ligands onto preset geometries so that parameters such as morphology, size, rigidity, and surface density map directly to the performance outputs of “loading-protection-release,” thereby providing a programmable framework for co-optimizing the carrier–cargo–biointerface [135]. In essence, these programmable parameters determine whether the structures can traverse the plant cell wall and release on demand inside cells, forming the key link that translates “material structure” into “delivery efficiency” [135].

In plant applications, DNA self-assembled structures have achieved efficient siRNA delivery and gene silencing in mature tissues, validating functional feasibility for these programmable nucleic-acid carriers in complex tissue environments [136]. Meanwhile, barrier effects associated with the cell wall and the endosome–vacuole pathway make combined optimization of morphology, charge, and surface ligands critical for improving tissue passability and intracellular release [136]. To ensure cross-platform reproducibility, plant-oriented standardized operating procedures now cover key steps including nucleic-acid complexation, tissue penetration, and silencing readouts, helping standardize the construction and use of origami/self-assembled carriers across materials and tissue levels [137].

Regarding physiological stability and biointerfaces, engineering coatings of sequence-defined peptoids (peptoid) on origami can markedly enhance resistance to nuclease degradation and optimize interactions with biomolecules, thereby expanding the usable window in complex fluid environments [138]. However, such stabilization strategies must still be co-designed with wall traversal and intracellular release in plant systems to further increase net delivery effect without adding physical damage [138].

Nonetheless, three core challenges remain for application: first, the physicochemical barrier of the plant cell wall limits internalization of nanostructures, and wall crossing and penetration mechanisms still require systematic elucidation; second, in vivo organelle-targeted delivery strategies for chloroplasts and mitochondria are not yet mature, with insufficient targeting precision; third, scalable manufacturing of complex DNA structures is constrained by synthesis efficiency, cost, and quality consistency, limiting large-scale and field scenarios. Looking ahead, efforts should focus on plant-oriented structural optimization (such as cell-wall permeation enhancement strategies coordinated with enzymatic/mechanical treatments), programmable organelle-targeting modules, and validation of delivery generalizability across species and tissues to unlock the high biocompatibility and modular potential of DNA nanostructures.

## 4. CRISPR-Cas Nano-Synergistic Editing Systems

CRISPR technology was first reported in 1987 [139], but its revolutionary potential was not fully realized until the past two decades. A key turning point came in 2012, when the Doudna–Charpentier team built a single-guide RNA-mediated Cas9 in vitro cleavage system, establishing a general editing foundation across species and application scenarios [140].

### 4.1. Common Vectors: CRISPR-Cas Is Hailed as “Molecular Scissors”

By targeting deletion, replacement, or editing of nucleic-acid sequences, it enables precise manipulation of plant genomes; core systems include family variants of Cas9, Cas12a, and Cas13a [141]. It should be emphasized that Cas9, with its mature PAM recognition and engineering ecosystem, occupies a dominant position in plant genetic improvement, whereas Cas13a offers the possibility of antiviral and reversible regulation through RNA targeting [142]. Trait-oriented work has edited rice EPFL9 for stomatal-trait improvement, validating the linkage from “editing → phenotype” [143]. Likewise, targeted mutations at tomato fruit-quality loci such as RIN/SlIAA9 demonstrate directed shaping of fruit traits [144]. In the same period, next-generation compact nucleases (such as Un1Cas12f1) achieved efficient editing in crops; their smaller size and easier packaging create structural conditions for delivery by viral vectors and nanocarriers, as validated by Tang et al. in rice and tomato [145]. Another handle is geminiviral replicons: the Botella team used CasΦ/Cas12f to demonstrate the synergistic potential of “small-payload editors × replicon amplification” [146].

#### 4.1.1. CRISPR-Cas DNA Vectors

Agrobacterium-mediated plasmid DNA delivery schemes, owing to their mature workflows, have been widely adopted in model plants and crops; early work achieved stable expression and transgenerationally heritable edited lines in tobacco and *Arabidopsis* [147]. On the crop side, the Gao Caixia team first accomplished CRISPR/Cas-targeted modifications in rice and wheat, establishing a feasible path from construct to whole plants (Figure 8A) [148]. Subsequently, Feng et al. and the Gao Caixia team carried out systematic evaluations of transgenerational inheritance and specificity in *Arabidopsis* and related systems, standardizing characterization paradigms [149]. To extend the application boundary, research has moved into hard-to-transform perennials such as fruit trees: Nishitani et al. at Japan’s NARO Institute of Fruit Tree Science completed editing in apples [150], and Ren et al. obtained targeted mutations in Chardonnay grape cells and plants [151]. It should be noted that DNA-vector approaches raise considerations of off-target effects and regulation due to random integration and persistent expression, underscoring the importance of DNA-free routes [152].

#### 4.1.2. Gene Gun Bombardment Delivery (RNP)

The RNP (the complex of Cas protein and sgRNA) approach can markedly reduce off-target effects and avoid residual foreign DNA fragments, thereby yielding non-transgenic (transgene-free) edited plants [152]. In monocot crops, wheat protoplast and immature embryo systems have achieved robust RNP editing and regeneration [153]. An important advance is that Yamada et al. directly delivered RNPs into zygotes of the maize inbred line B73 and regenerated DNA/marker double-negative plants, providing a credible paradigm for tissue culture-free and precision breeding (Figure 8B) [154]. In line with this, the transient nature of RNPs requires coordinated design for protein stability, cellular uptake, and nuclear delivery to prevent loss of activity and an overly short editing window [153].

#### 4.1.3. Viral/Nano-Synergistic Vectors and Organelle Targeting

At the level of “organelle editing,” in vivo targeted delivery to chloroplast and mitochondrial genomes remains a major challenge [141]. In parallel, the Kwak–Giraldo–Strano team proposed a CNT–chloroplast targeted-delivery framework and achieved expression within chloroplasts across multiple plants, providing a co-design concept of materials and ligands for organelle-targeted editing [80]. At the heritable level, Liu et al. realized germline-heritable VIGE in latent axillary meristems of tomato (Figure 8C) [155]; the Staskawicz–Dinesh–Kumar team used TRV-gRNA (carrying a mobile RNA sequence) to obtain high-frequency germline editing [156]. Mechanistically, these advances, combined with the loading advantages of compact nucleases, assemble modular building blocks for “compact editors × viral/nano-delivery × germline/organ targeting.”

**Figure 8 plants-14-03625-f008:**
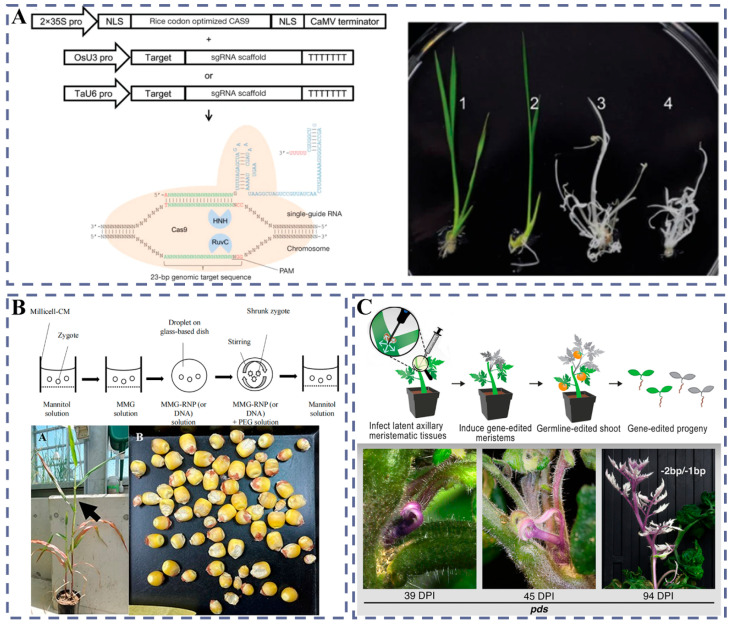
(**A**) Schematic of the engineered type II CRISPR-Cas system. Cas9’s HNH and RuvC-like domains cleave the target sequence’s strands when the correct PAM sequence is present. NLSs, nuclear localization signals. Reprinted with permission from ref. [148], Copyright 2013, Springer Nature. (**B**) PEG-mediated delivery of Cas9-gRNA RNPs to maize-B73 zygotes and verification of mutagenesis in regenerated plants; reprinted with permission from ref. [154], Copyright 2024, Oxford University Press. (**C**) Generation of gene-edited shoots via viral infection of dormant axillary meristematic cells. Reprinted with permission from Ref. [155], Copyright 2024.

### 4.2. Editing-Efficiency Advantages and Development Potential

Nanocarriers can provide multi-stage gains by improving component stability, lowering nuclease degradation, enhancing cellular uptake, and promoting endosomal/vacuolar escape, thereby jointly expanding the effective editing window of CRISPR systems [157]. More critically, when combined with tissue culture-free regeneration strategies, it becomes possible to realize an integrated “construct-deliver-regenerate” workflow in crops, shortening the trait-development cycle and lowering barriers to entry [1]. Methodologically, Qiu et al. showed that direct mRNA delivery can drive efficient editing in plants while avoiding risks of DNA integration, facilitating interfacing with nano/viral vectors [158]. On the crop side, Liu et al. significantly increased prime editing efficiency in rice and broadened the editable site spectrum by conditionally downregulating *OsMLH1* [159]. Another route is prime editing (PE): the Zong/Ni team, using ePPEplus in hexaploid wheat, raised efficiency to levels enabling multi-locus/multigene editing, providing a higher ceiling for complex replacements and fine-tuning [160]. Going forward, development should integrate the latest engineering consensus on precise nano-delivery in plants and pursue systems co-designed around “editor size-packaging mode-tissue/germline localization-organelle targeting-regeneration route,” to drive nano-synergistic precision editing toward application [161].

## 5. Rational Design of Nanocarriers and Multiscale Delivery

The rational design of nanocarriers for plant genetic engineering must simultaneously solve two coupled problems—how to reach the right biological address across tissues, cell types, and organelles with high specificity, and how to do so in a manner that is environmentally sustainable, degradable, and compatible with regulatory expectations. Integrating these dimensions requires a life-cycle perspective that links material structure, surface chemistry, and stimulus responsiveness to transport across the plant surface, cell wall, membranes, and subcellular barriers, while also anticipating environmental aging, eco-corona formation, dissolution, enzymatic degradation, and biotic interactions in soil–plant–microbe systems. In practice, design should combine passive navigation (size, aspect ratio, surface charge, wettability) with active recognition (ligands for membrane transporters and organelle receptors), and encode release logic that is matched to plant microenvironments. Recent field-wide consensus emphasizes that precision delivery in plants benefits from co-optimizing carrier geometry and charge with receptor-mediated recognition, and from reporting not just mass dose but also number concentration, surface area, and effective deposition per leaf area, to enable reproducible comparisons across species and growth stages [91,161,162]. To ensure reproducibility and translational relevance, we align this section with the structure and terminology of the present manuscript and its materials typology, while expanding on targeting strategies, verification workflows, and environmental fate to form a coherent “design-deliver-degrade” narrative.

At the targeting level, chloroplasts and mitochondria are the most mature subcellular destinations for active delivery. Chitosan-complexed single-walled carbon nanotubes (SWNTs) demonstrate chloroplast-selective delivery and gene expression in planta, indicating that pairing a high-aspect-ratio scaffold with polycationic shells can achieve organelle selectivity without cell-wall removal [80]. Complementarily, clustered cell-penetrating peptides fused with chloroplast-targeting or mitochondria-targeting sequences can deliver plasmid DNA to diverse plastid types and even enable targeted integration into organelle genomes, establishing a modular “peptide address label” strategy that generalizes across tissues [129,130]. Chemical biorecognition motifs have also enabled nanomaterials to traffic with chemical cargoes to specific tissues such as photosynthetic organs, highlighting that ligand–receptor or membrane–component interactions can be designed into the carrier surface to raise local enrichment and functional readouts [60]. Intriguingly, for RNA interference in mature leaves, nanoparticle internalization is not strictly required to achieve gene silencing: gold nanorods that enter and 10 nm gold spheres that remain extracellular can yield inverted silencing outcomes, decoupling “cellular entry” from “functional effect” and expanding the design space toward surface-confined, extracellular pathways that exploit apoplast flow and cell-to-cell transport [107]. At the organ and whole-plant levels, phloem-directed transport can be enhanced by sucrose-related recognition elements, enabling leaf-to-root translocation and improving long-distance delivery after foliar application, whereas layered double hydroxide (LDH) platforms and their dsRNA “BioClay” formulations enable durable retention and controlled release on leaf surfaces, providing 20–30 day protection windows and rapid in-leaf uptake of functional siRNA with >70% mRNA reduction in intact tissues [98,105]. Within the mesoporous silica nanoparticle (MSN) paradigm, pore encapsulation with labile end-caps (for example, gold-nanoparticle caps) and stimulus-responsive gates supports “leak-proof” loading plus triggered release; recent studies have achieved long-term, multigene silencing by simple foliar dosing, underscoring how synthesis-stage surface functionalization and ligand grafting translate into plant-level specificity and durability [95,97]. Stimuli-responsive peptide carriers add a second, orthogonal layer of control by coupling pH/reduction sensitivity to endosomal escape and intracellular release, thereby improving the probability that nucleic acids reach the nucleus or plastid matrix intact and active [128]. DNA nanostructures—through programmed geometry, stiffness, and ligand display—further allow wall traversal and siRNA coordination to be tuned with sub-10 nm precision; peptoid coatings stabilize these structures against nuclease attack while providing chemical handles for attaching targeting ligands, enabling a continuum from extracellular anchoring to subcellular targeting [136,137,138]. Ultrasmall cationic gold nanoclusters (~2 nm) illustrate a complementary chemical self-delivery route that protects siRNA, enters mature leaves by infiltration, and yields strong silencing with tunability via cluster size and polyethylenimine (PEI) molecular weight. Meanwhile, high-aspect-ratio CNT scaffolds enable non-integrative gene expression across species and can be layered with ligands to gain organelle or tissue specificity [61,79,163]. Across these platforms, a pragmatic synthesis-stage workflow is to pre-assemble a navigation layer (e.g., sugar motifs, chloroplast transit peptides), a transport layer (e.g., high aspect ratio or sub-10 nm clusters with tuned ζ-potential and hydration), and a release layer (e.g., pH/reduction-responsive gates on MSNs or cleavable peptide domains), and to quantify ligand density and surface heterogeneity alongside size distribution, ζ-potential, and polydispersity to ensure batch-to-batch consistency [95,97,162].

Verification should minimize false-positive “apparent targeting.” A rigorous sequence is to first quantify spatial biodistribution at the organ and tissue scales with whole-leaf imaging or ICP-MS, then validate cell/organellar co-localization and function using marker lines and qPCR/protein readouts, and finally confirm specificity via competition or receptor-loss controls. For instance, lowering phloem loading by adding sucrose transporter competitors, or ablating plastid targeting by mutating transit-peptide motifs [164,165]. In reporting, delivery metrics should include per-area deposition, number-based dose, and surface area dose, together with dispersion state and aging in the actual exposure medium, because electrostatics and wetting govern interactions with the leaf surface, stomatal aperture, and the cell wall’s cellulose–hemicellulose–pectin network [91,162]. This quantification connects directly to rational synthesis: electrostatic attraction can improve initial adhesion yet also increases non-specific binding. Thus, zwitterionic or glycosylated brushes can be used to tune non-specific adsorption without compromising ligand-receptor recognition.

The environmental dimension should be elevated to a first-class design constraint. International guidance from EFSA and OECD articulates how to characterize small-particle fractions, dissolution, exposure, and hazard, and to standardize sample preparation and dosimetry for nanomaterials; these frameworks map well onto plant nano-delivery because they require reporting of dispersion state, surface charge, aging, and dose metrics across mass, number, and surface area [166,167]. In soil–plant systems, carbon nanomaterials exemplify the possibility of enzymatic biodegradation, together with context-dependent ecological effects: peroxidases such as myeloperoxidase and horseradish peroxidase can oxidatively shorten CNTs over weeks, reducing persistence, whereas in soils, CNTs and graphene derivatives can agglomerate, age, and shift bacterial community structure and nitrogen cycling in dose-, functionalization-, and soil-type-dependent ways [168,169,170,171]. MSNs present a contrasting case: they gradually hydrolyze to orthosilicic acid, providing a chemically defined dissolution path with generally low acute toxicity, although cationic surfaces and high doses can elicit irritation; controlled pore geometry and surface amination govern dissolution kinetics and uptake, so reporting time-resolved silicic-acid release is critical for environmental assessment [172,173]. LDH carriers combine anion-exchange-mediated loading with partial dissolution and metal-ion release; Mg-Al LDH nanosheets and nanoparticles have shown low acute toxicity in common concentration windows in plant and fungal systems, yet dopants (e.g., Cu/Co) can increase algal toxicity, underscoring the need to tune metal composition and quantify ion release under foliar and soil conditions [174,175]. Noble-metal platforms are generally chemically inert and therefore environmentally persistent; gold nanoparticles and clusters often show low acute toxicity but can bioaccumulate and induce subtle biochemical responses in plants and soil invertebrates, making it prudent to minimize metal mass, favor ultrasmall clusters with clearable ligands, and assess leaching and trophic transfer in realistic soil microcosms [176,177]. Liposomal and other phospholipid carriers are readily cleaved by plant and microbial phospholipases A2/C/D into glycerophosphates and fatty acids, providing good environmental degradability; formulation variables such as cholesterol content and stabilizers modulate leaf retention and thus exposure duration [178,179]. Chitosan-based systems, broadly degradable by chitinases and lysozyme, typically show low ecological risk and can even modulate beneficial plant defenses, though quaternization or PEGylation can alter degradation and should be evaluated explicitly [180,181]. DNA and peptide carriers are, in principle, rapidly degraded by nucleases and proteases. However, stabilization strategies that extend functional windows—such as peptoid coatings on DNA origami—also extend persistence, so plant-context co-design must balance efficacy with biodegradability to avoid undue accumulation [138,182,183]. For sprayable dsRNA systems, BioClay illustrates how “surface residency plus controlled release” can deliver durable protection while ultimately degrading; quantitative residue and desorption kinetics on field-grown leaves should accompany efficacy claims to support exposure modeling [105].

Bringing these strands together, a unifying design principle is to encode specificity and degradability at the synthesis stage and to validate them at multiple biological scales. On the specificity side, ligand-guided targeting to plastids, mitochondria, vasculature, or leaf surfaces can be layered onto geometries that cross the wall or lodge at the epidermal–apoplast interface; stimulus-responsive gates or peptides then couple entry to intracellular release and endosomal/vacuolar escape. On the sustainability side, preference should be given to carriers with clear and benign degradation or dissolution paths, such as MSNs to orthosilicic acid, LDH with tunable metal chemistry and exchange, lipidic and polysaccharide carriers degradable by endogenous enzymes, and DNA or peptide assemblies that are nucleolytically or proteolytically cleared within relevant time frames [184,185,186,187]. The design envelope can also accommodate extracellular silencing routes that minimize the need for cellular internalization, reducing long-term intracellular persistence, as shown by decoupled entry/effect in gold systems and by LDH-based surface residency with functional siRNA uptake [98,188]. Across all platforms, quantitative reporting of dose metrics, dispersion/aging, and residue kinetics in realistic plant and soil media, together with organ-to-organ transport and whole-plant biodistribution, should be treated as core data products rather than optional add-ons, enabling comparisons across species and environments and smoothing the path to regulatory acceptance [189,190,191]. With such a “design-deliver-degrade” framework, nanocarrier-enabled plant genetic engineering can move toward precise, tissue-aware, and environmentally responsible applications at field scale.

## 6. Summary and Prospect

The evolution of nanotechnology in plant genetic engineering has ushered in fresh optimism for the genetic enhancement of crops. Conventional techniques, such as Agrobacterium-mediated transformation and the gene gun, are frequently hampered by issues like species-specific limitations, tissue damage, and the stochastic integration of exogenous DNA. In contrast, nanomaterials offer a more adaptable and less invasive approach. The minute size and customizable nature of diverse nanocarriers, including carbon-based nanomaterials, inorganic nanoparticles, and liposomes, allow them to penetrate the rigid plant cell wall and directly introduce genetic materials like DNA, RNA, or proteins into the cell’s interior. This “intelligent delivery” method not only minimizes harm to plant tissues but also circumvents the permanent integration of foreign genes into the plant genome, enabling non-transgenic transient expression and precise regulation. Crucially, nanocarriers have demonstrated the potential to work in concert with precise gene-editing tools like CRISPR: by efficiently delivering CRISPR/Cas components to specific sites within plant cells, they can potentially enhance the success rate of gene editing and enable targeted modifications of specific tissues or organelles. It can be said that nanotechnology provides a programmable toolkit for plant genetic engineering, making precise operations that were previously challenging now feasible.

Nevertheless, these advantages must be balanced against limitations including material heterogeneity and batch-to-batch variability, incomplete understanding of long-term environmental fate and food-chain transfer, and the possibility of off-target physiological or microbiome effects.

Of course, to fully leverage the potential of nanocarriers in plants, we still face numerous challenges. One key issue is the optimization of delivery efficiency and targeting. Currently, different plant species and tissues respond variably to nanocarriers: a method that achieves high efficiency in model plants may not work well in other crops or mature tissues. This is due to differences in cell wall structure, cell types, and physiological environments among plant species, which affect the absorption and transport of nanoparticles. Therefore, improving delivery efficiency means tailoring the size, surface properties, and functional design of carriers to different contexts to ensure that more genetic cargo can cross barriers and accurately reach target cells. Meanwhile, targeting is also a direction for future efforts—how to release the gene-editing tools carried by nanocarriers at the right time and place and specifically act on the cell types that need editing is a problem yet to be solved. To this end, researchers are exploring strategies such as adding bio-targeting molecules and controlled-release mechanisms to make gene delivery more selective and controllable. In addition, in terms of cross-species applications, we need to verify and adjust the universality of these technologies. Ideally, a nano-delivery system should be applicable to a variety of crops and different organs (from leaves to roots, and even pollen or embryos), which requires us to continuously accumulate data in practice and improve the adaptation of nanocarriers to different plants. Beyond efficiency and targeting, reproducibility remains a practical constraint: differences in composition, surface chemistry, and manufacturing often lead to variable biological responses; standardized dose metrics (e.g., μg cargo per cm^2^ leaf or μg/mL for infiltration), application volume/exposure time, and required controls/replicates should be consistently reported to enable cross-lab comparison. Potential phytotoxicity, residual accumulation, and microbiome perturbation warrant routine testing under greenhouse and field settings, together with life-cycle and degradability assessments. Finally, scale-up and regulatory readiness are non-trivial, encompassing cost, quality control, and alignment with environmental and food-safety frameworks. Although these challenges are real, they are not insurmountable, but rather point the way for further engineering improvements.

## Data Availability

No new data were created or analyzed in this study. Data sharing is not applicable to this article.

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
