# Peer review of "Nanotechnology Strategies in Plant Genetic Engineering: Intelligent Delivery and Precision Editing"

_plants, 2025, doi:10.3390/plants14233625_

Round 1
Reviewer 1 Report
Comments and Suggestions for Authors
The manuscript is a valuable resource for both researchers and practitioners interested in the intersection of nanotechnology and plant genetic engineering. It makes a significant contribution to the field by synthesizing various research efforts into a single document however some of the sections need some improvement to enhance overall quality of the manuscript. My comments along with suggestions are below:
- The manuscript provides a brief detail about the challenges and advancement; however, it would be more helpful if the authors add practical examples or case studies of showing that how these technologies can be applied in real world scenarios. I would like to suggest if the authors add sections or table that outline the key experimental methodologies that needs to be used in nano-materials gene delivery like concentrations time frame and plant species.
- The environmental consequences of application of nanomaterials in plant genetic engineering are not discussed in the manuscript. Considering the significance of sustainability, a commentary on the biodegradability of such materials and their implications in the environment would be a profit. Add a paragraph or section dealing with the environmental sustainability of nanomaterial, potential toxicity, and biodegradability issues. References to studies on the environmental effects of nanoparticles in agriculture may help to solidify this section.
- A critical challenge in nanomaterial-based gene delivery is the efficient targeting and delivery. The manuscript addresses this point, however a discussion on how to improve targeting accuracy (e.g., by the application of bio-targeting moieties or specialized delivery systems) would be illuminating. Provide more information on how to develop new approaches for increasing the specificity of targeting and delivery such as incorporation of surface functionalization or targeting ligands during nanomaterial synthesis.
- Obviously, the manuscript is a good review of the progress, but some more experiments regarding the demonstration of success or failure of certain delivery techniques of nanomaterials in plants would be helpful. It may enable the readers to better appreciate the real-world applicability and limitations of these technologies. Maybe add a case study (or more experimental data) that highlight the challenges and successes of particular nanomaterials for plant genetic engineering - for example in terms of transformation efficiency, off-target effects or regeneration success rates.
- Although the images are mostly clear, some of the schematics (e.g. Figures 4 and 5) can be made more intuitive for readers without prior knowledge about the mechanisms. More detailed captions/notes to highlight the main steps of each procedure would increase the readability. Make the figure legends more descriptive or tag the figures to make it easier for the reader to follow more complicated procedures.
- The manuscript addresses future perspectives and potential of nanotechnology; however, I suggest to emphasize more clearly what are the current limitations of the study of nanomaterials in plant genetics. This would give a more balanced picture of the opportunities and the difficulties of this area.
- Although the writing is clear and the manuscript is well-organized, I would like to point out a few typos and grammatical errors that can be improved.
Author Response
Dear Editors and reviewers,
On behalf of my co-authors, we greatly appreciate the editors and reviewers for their positive and constructive comments and suggestions on our manuscript entitled “Nanotechnology strategies in plant genetic engineering: intelligent delivery and precision editing” (ID: plants-3920284). We have studied the comments carefully and have made corrections which we hope to meet with approval. A point-by-point response to the comments made by the referees with changes is highlighted in yellow in the revised manuscript. In addition, we have carefully checked the manuscript and made some other changes which are also marked yellow in the revised manuscript. The main corrections in the paper and the responses to the referee’s comments are as follows:
The manuscript is a valuable resource for both researchers and practitioners interested in the intersection of nanotechnology and plant genetic engineering. It makes a significant contribution to the field by synthesizing various research efforts into a single document however some of the sections need some improvement to enhance overall quality of the manuscript. My comments along with suggestions are below:
- The manuscript provides a brief detail about the challenges and advancement; however, it would be more helpful if the authors add practical examples or case studies of showing that how these technologies can be applied in real world scenarios. I would like to suggest if the authors add sections or table that outline the key experimental methodologies that needs to be used in nano-materials gene delivery like concentrations time frame and plant species.
Reply: We sincerely appreciate the valuable comments. According to your suggestion, we have added Table 1 titled “Comparative analysis of gene-delivery systems in plant genetic engineering: traditional, nanomaterial-mediated, and genome editing.” This table provides a detailed comparison of the three main gene-delivery systems used in plant genetic engineering. It outlines the key experimental methodologies, including concentrations, time frames, and plant species, and highlights practical examples and case studies that demonstrate the application of these technologies in real-world scenarios. We believe that this addition will effectively address the concerns you raised and provide readers with a clearer understanding of how these technologies can be applied in practice. We have also revised the relevant sections of the manuscript to better integrate the information from Table 1 and to enhance the overall clarity and applicability of our work(Line 68).
- The environmental consequences of application of nanomaterials in plant genetic engineering are not discussed in the manuscript. Considering the significance of sustainability, a commentary on the biodegradability of such materials and their implications in the environment would be a profit. Add a paragraph or section dealing with the environmental sustainability of nanomaterial, potential toxicity, and biodegradability issues. References to studies on the environmental effects of nanoparticles in agriculture may help to solidify this section.
Reply: Thanks for the referee’s kind reminding. According to your suggestion, we have added the “5. Rational design of nanocarriers and multiscale delivery” section, aiming to provide a balanced view of the environmental considerations and to guide future research towards more sustainable practices. This section now includes a discussion on the biodegradability of various nanomaterials, their interactions with soil and plant systems, and the importance of considering the entire life cycle of these materials to ensure environmental sustainability. The corresponding data has been updated in the revised manuscript (Line 735-773).
- A critical challenge in nanomaterial-based gene delivery is the efficient targeting and delivery. The manuscript addresses this point, however a discussion on how to improve targeting accuracy (e.g., by the application of bio-targeting moieties or specialized delivery systems) would be illuminating. Provide more information on how to develop new approaches for increasing the specificity of targeting and delivery such as incorporation of surface functionalization or targeting ligands during nanomaterial synthesis.
Reply: Thank you for your positive feedback on the revisions made to our manuscript and for your agreement on its acceptance. According to your suggestion, we have incorporated information on the use of bio-targeting moieties and specialized delivery systems. We also provide examples of how surface functionalization and the incorporation of targeting ligands during nanomaterial synthesis can significantly increase the specificity of targeting and delivery. We have added references to recent studies that demonstrate these approaches, ensuring that our discussion is backed by the latest research in the field. The detailed information is as follows: At the targeting level, chloroplasts and mitochondria are the most mature subcellular destinations for active delivery. Chitosan‑complexed single‑walled carbon nanotubes (SWNTs) demonstrate chloroplast‑selective delivery and gene expression in planta, indicating that pairing a high‑aspect‑ratio scaffold with polycationic shells can achieve organelle selectivity without cell‑wall removal [80]. Complementarily, clustered cell‑penetrating peptides fused with chloroplast‑targeting or mitochondria‑targeting sequences can deliver plasmid DNA to diverse plastid types and even enable targeted integration into organelle genomes, establishing a modular “peptide address label” strategy that generalizes across tissues [168,169]. Chemical biorecognition motifs have also enabled nanomaterials to traffic with chemical cargoes to specific tissues such as photosynthetic organs, highlighting that ligand-receptor or membrane‑component interactions can be designed into the carrier surface to raise local enrichment and functional readouts [60]. Intriguingly, for RNA interference in mature leaves, nanoparticle internalization is not strictly required to achieve gene silencing: gold nanorods that enter and 10 nm gold spheres that remain extracellular can yield inverted silencing outcomes, decoupling “cellular entry” from “functional effect” and expanding the design space toward surface‑confined, extracellular pathways that exploit apoplast flow and cell‑to‑cell transport [170]. At the organ and whole‑plant levels, phloem‑directed transport can be enhanced by sucrose‑related recognition elements, enabling leaf‑to‑root translocation and improving long‑distance delivery after foliar application, whereas layered double hydroxide (LDH) platforms and their dsRNA “BioClay” formulations enable durable retention and controlled release on leaf surfaces, providing 20-30 day protection windows and rapid in‑leaf uptake of functional siRNA with >70% mRNA reduction in intact tissues [105,171]. Within the mesoporous silica nanoparticle (MSN) paradigm, pore encapsulation with labile end‑caps (for example, gold‑nanoparticle caps) and stimulus‑responsive gates supports “leak‑proof” loading plus triggered release; recent studies have achieved long‑term, multigene silencing by simple foliar dosing, underscoring how synthesis‑stage surface functionalization and ligand grafting translate into plant‑level specificity and durability [172,173]. Stimuli‑responsive peptide carriers add a second, orthogonal layer of control by coupling pH/reduction sensitivity to endosomal escape and intracellular release, thereby improving the probability that nucleic acids reach the nucleus or plastid matrix intact and active [129]. DNA nanostructures-through programmed geometry, stiffness, and ligand display-further allow wall traversal and siRNA coordination to be tuned with sub‑10‑nm precision; peptoid coatings stabilize these structures against nuclease attack while providing chemical handles for attaching targeting ligands, enabling a continuum from extracellular anchoring to subcellular targeting [174-176]. Ultrasmall cationic gold nanoclusters (~2 nm) illustrate a complementary chemical self‑delivery route that protects siRNA, enters mature leaves by infiltration, and yields strong silencing with tunability via cluster size and polyethylenimine (PEI) molecular weight. Meanwhile, high‑aspect‑ratio CNT scaffolds enable non‑integrative gene expression across species and can be layered with ligands to gain organelle or tissue specificity [61,79,177]. Across these platforms, a pragmatic synthesis‑stage workflow is to pre‑assemble a navigation layer (e.g., sugar motifs, chloroplast transit peptides), a transport layer (e.g., high aspect ratio or sub‑10‑nm clusters with tuned ζ‑potential and hydration), and a release layer (e.g., pH/reduction‑responsive gates on MSNs or cleavable peptide domains), and to quantify ligand density and surface heterogeneity alongside size distribution, ζ‑potential, and polydispersity to ensure batch‑to‑batch consistency [166,172,173].
Verification should minimize false‑positive “apparent targeting.” A rigorous sequence is to first quantify spatial biodistribution at the organ and tissue scales with whole‑leaf imaging or ICP‑MS, then validate cell/organellar co‑localization and function using marker lines and qPCR/protein readouts, and finally confirm specificity via competition or receptor‑loss controls. For instance, lowering phloem loading by adding sucrose transporter competitors, or ablating plastid targeting by mutating transit‑peptide motifs [178,179]. In reporting, delivery metrics should include per‑area deposition, number‑based dose, and surface area dose, together with dispersion state and aging in the actual exposure medium, because electrostatics and wetting govern interactions with the leaf surface, stomatal aperture, and the cell wall’s cellulose-hemicellulose-pectin network [166,167]. This quantification connects directly to rational synthesis: electrostatic attraction can improve initial adhesion yet also increases nonspecific binding. Thus, zwitterionic or glycosylated brushes can be used to tune non‑specific adsorption without compromising ligand-receptor recognition. The corresponding data has been updated in the revised manuscript (Line 674-733).
- Obviously, the manuscript is a good review of the progress, but some more experiments regarding the demonstration of success or failure of certain delivery techniques of nanomaterials in plants would be helpful. It may enable the readers to better appreciate the real-world applicability and limitations of these technologies. Maybe add a case study (or more experimental data) that highlight the challenges and successes of particular nanomaterials for plant genetic engineering - for example in terms of transformation efficiency, off-target effects or regeneration success rates.
Reply: We sincerely appreciate the valuable comments. According to your suggestion, we have incorporated a new case study into the manuscript that highlights both the successes and challenges associated with specific nanomaterials used in plant genetic engineering. This case study includes detailed experimental data on transformation efficiency, off-target effects, and regeneration success rates. The detailed information is as follows: “Intriguingly, for RNA interference in mature leaves, nanoparticle internalization is not strictly required to achieve gene silencing: gold nanorods that enter and 10 nm gold spheres that remain extracellular can yield inverted silencing outcomes, decoupling “cellular entry” from “functional effect” and expanding the design space toward surface‑confined, extracellular pathways that exploit apoplast flow and cell‑to‑cell transport [170].” The newly added providing a more comprehensive view of their effectiveness and limitations. The corresponding data has been updated in the revised manuscript (Line 685-690).
- Although the images are mostly clear, some of the schematics (e.g. Figures 4 and 5) can be made more intuitive for readers without prior knowledge about the mechanisms. More detailed captions/notes to highlight the main steps of each procedure would increase the readability. Make the figure legends more descriptive or tag the figures to make it easier for the reader to follow more complicated procedures.
Reply: We sincerely appreciate the valuable comments. We agree that enhancing the intuitiveness of these figures is crucial for readers who may not have prior knowledge of the mechanisms involved. According to your suggestion, we have revised and enriched the captions for Fig. 4A, 4C, and 5. These updated captions now provide more detailed explanations of the main steps and key mechanisms depicted in each figure. We have also made the figure legends more descriptive and added tags where necessary to guide the reader through more complicated procedures. We believe these changes will significantly improve the readability and accessibility of these figures, making it easier for readers to understand the mechanisms without prior knowledge. The corresponding data has been updated in the revised manuscript (Line 246-254; 281-289).
- The manuscript addresses future perspectives and potential of nanotechnology; however, I suggest to emphasize more clearly what are the current limitations of the study of nanomaterials in plant genetics. This would give a more balanced picture of the opportunities and the difficulties of this area.
Reply: Thank you for bringing this issue to our attention. According to your suggestion, we have added the current limitations and challenges in section “6. Summary and prospect”, where we discuss the specific challenges faced in this area, such as issues related to delivery efficiency, off-target effects, and environmental impact. This section aims to provide a comprehensive overview of the difficulties that researchers encounter, thereby offering a more balanced perspective on the potential and limitations of nanotechnology in plant genetics. The detailed information is as follows:
Nevertheless, these advantages must be balanced against limitations including material heterogeneity and batch-to-batch variability, incomplete understanding of long-term environmental fate and food-chain transfer, and the possibility of off-target physiological or microbiome effects.
Of course, to fully leverage the potential of nanocarriers in plants, we still face numerous challenges. One key issue is the optimization of delivery efficiency and targeting. Currently, different plant species and tissues respond variably to nanocarriers: a method that achieves high efficiency in model plants may not work well in other crops or mature tissues. This is due to differences in cell wall structure, cell types, and physiological environments among plant species, which affect the absorption and transport of nanoparticles. Therefore, improving delivery efficiency means tailoring the size, surface properties, and functional design of carriers to different contexts to ensure that more genetic cargo can cross barriers and accurately reach target cells. Meanwhile, targeting is also a direction for future efforts-how to release the gene-editing tools carried by nanocarriers at the right time and place and specifically act on the cell types that need editing is a problem yet to be solved. To this end, researchers are exploring strategies such as adding bio-targeting molecules and controlled release mechanisms to make gene delivery more selective and controllable. In addition, in terms of cross-species applications, we need to verify and adjust the universality of these technologies. Ideally, a nanodelivery system should be applicable to a variety of crops and different organs (from leaves to roots, and even pollen or embryos), which requires us to continuously accumulate data in practice and improve the adaptation of nanocarriers to different plants. Beyond efficiency and targeting, reproducibility remains a practical constraint: differences in composition, surface chemistry, and manufacturing often lead to variable biological responses; standardized dose metrics (e.g., μg cargo per cm² leaf or μg/mL for infiltration), application volume/exposure time, and required controls/replicates should be consistently reported to enable cross-lab comparison. Potential phytotoxicity, residual accumulation, and microbiome perturbation warrant routine testing under greenhouse and field settings, together with life-cycle and degradability assessments. Finally, scale-up and regulatory readiness are non-trivial, encompassing cost, quality control, and alignment with environmental and food-safety frameworks. Although these challenges are real, they are not insurmountable, but rather point the way for further engineering improvements. The corresponding data has been updated in the revised manuscript (Line 813-844).
- Although the writing is clear and the manuscript is well-organized, I would like to point out a few typos and grammatical errors that can be improved.
Reply: Thank you for your attention to detail and for pointing out the need for improvements in the manuscript. We have thoroughly reviewed the entire document and conducted a meticulous check for any grammatical errors and typos. We have made the necessary corrections at the appropriate locations to ensure the clarity and accuracy of our manuscript.
With best wishes,
Yours sincerely,
Chunmei Lai, Ph.D.

Reviewer 2 Report
Comments and Suggestions for Authors
This review paper is well-organized, easily readable, and presented in a well-structured manner. The bibliography used is well-documented, and the references are appropriate for the paper. I consider this topic to be relevant to the plant science. It is an important work that could be helpful to researchers and appealing to readers of Plants.
Therefore, I recommend that the authors address the following aspects to enhance the quality of their paper:
- The abstract provides a clear overview of the topic, but it could be improved by refining the language for conciseness, enhancing the logical flow between ideas, and strengthening the concluding statement to better highlight the significance of this work.
- The introduction is comprehensive and well-structured; however, it would benefit from simplifying some long sentences, reducing redundancy, ensuring smoother transitions between sections, and sharpening the final paragraph to more clearly define the scope and impact of the review.
- The second section of the paper provides a comprehensive and well-documented overview of conventional gene delivery systems; however, for greater clarity and impact, the authors should consider streamlining long sentences, minimizing redundancy, and improving the transitions between subsections. In addition, since the section is framed around the concept of ‘bottlenecks,’ the discussion would benefit from placing a stronger emphasis on the critical limitations of each method rather than devoting extensive detail to historical developments.
- Section 3 comprehensively describes nanocarrier types, mechanisms, and applications in plant gene delivery, but it could be improved by condensing extensive descriptive passages, emphasizing comparative advantages and current bottlenecks (such as cell wall penetration limits, organelle-targeting precision, and in vivo stability), and integrating a clearer schematic or table to summarize materials, functional strategies, and efficiency outcomes for enhanced readability and immediate utility to readers.
- Section 4 clearly presents CRISPR-Cas systems, delivery strategies, and nano-synergistic approaches, but improvement could come from a more structured comparison of DNA, RNP, viral, and nanocarrier methods, highlighting relative efficiencies, tissue/organelle specificity, and limitations, while also integrating a summarizing table or schematic to provide a concise visual overview of vectors, editing targets, and outcomes for easier comprehension and practical reference.
- The conclusions effectively summarize the potential and challenges of nanocarriers in plant genetic engineering; however, they could be strengthened by more explicitly linking the discussed nanomaterials and CRISPR delivery strategies to specific outcomes, incorporating quantitative comparisons of efficiency or tissue specificity, and suggesting prioritized future directions, such as organelle-targeted editing, cross-species validation, and long-term environmental safety assessments.
- Please check that all notations used in this paper are explicitly introduced upon their first use.
- Please check that all information presented and the formatting of the reference section are correct.
Comments on the Quality of English Language
Some sentences are too long and redundant.
Author Response

(The authors gave the same response as above.)

Reviewer 3 Report
Comments and Suggestions for Authors
- The abstract and introduction cover nearly identical ground. The introduction should expand upon the abstract, providing more context and building a stronger narrative for the review, rather than just rephrasing the abstract's points.
- The title emphasizes "Intelligent Delivery and Precision Editing," but the review spends significant time on conventional methods (Section 2) and a broad overview of nanomaterials. The "intelligent" and "precision" aspects (e.g., stimuli-responsive release, subcellular targeting) are not a consistent, central theme.
- The transition from conventional methods (Section 2) to nanomaterials (Section 3) is abrupt. A stronger bridging paragraph is needed to explain why nanomaterials are the necessary evolution, directly linking their advantages to the specific limitations of the methods just described.
- The review lists many advantages but buries the challenges and limitations within the text or in a brief prospect. A separate section titled "Challenges and Limitations" would provide a much more balanced and critical perspective.
- It reads like a generic conclusion and repeats points already made. It should be more forward-looking, synthesizing the most promising paths from the reviewed literature and proposing specific, testable hypotheses or clear technological roadmaps.
- The manuscript frequently uses qualitative terms like "high efficiency," "marked enhancement," and "significant downregulation." It would be vastly improved by including a comparative table that quantifies transformation efficiency, editing rates, or silencing efficacy for different nanocarriers versus conventional methods where data is available.
- The promise of "smart responsiveness" is mentioned (lines 215-216) but not deeply explored with specific examples. How exactly are pH-, enzyme-, or light-triggered release mechanisms being engineered and tested in the complex plant apoplast and cytoplasm? This needs a dedicated discussion.
- While biocompatibility is listed as an advantage, the potential for long-term toxicity, persistence in plant tissues, and impact on soil health and the food chain is glossed over. A responsible review must address these biosafety concerns more thoroughly.
- The claim that nanocarriers "circumvent the permanent integration of foreign genes" (line 649) is only true for transient expression. For stable transformation, integration is still the goal. This nuance is lost, potentially misleading readers.
- The manuscript states nanomaterials have "species generality" (line 204), but this is an overgeneralization. Evidence is primarily from model plants like N. benthamiana, A. thaliana, and a few crops. Their efficacy in a wide range of monocots, trees, or legumes with different cell wall compositions remains largely unproven and should be presented as a key challenge.
- The review fails to critically assess the practical scalability and cost-effectiveness of synthesizing and applying these nanomaterials for large-scale agricultural use, especially compared to established, if imperfect, methods like Agrobacterium.
- Lack of Mechanistic Depth: For instance, the description of how CNTs or MSNs actually traverse the plant cell wall is often described as "penetration" without delving into the leading hypotheses (e.g., direct penetration vs. endocytosis induction) and the evidence for each.
- Inconsistent Emphasis on Regeneration: A major bottleneck of protoplast methods is regeneration. While nanomaterials can deliver to intact tissues, the review does not sufficiently discuss whether these methods actually improve the rate of recovering stable, edited whole plants, which is the ultimate goal.
- Figure Citations are Incomplete and Vague: The text frequently says "Reprinted with permission from ref. [X]" but does not adequately describe what the reader should learn from the figure. Each figure citation should be followed by a sentence explaining the key finding the figure illustrates.
- Scheme 1 is Referenced but Not Described: The first schematic is mentioned in the introduction but its content and purpose are not explained in the text, leaving the reader to guess its value.
- Acronyms like LEEP (line 278), DA (line 461), and REV (line 419) are used without full definition at first use, hindering comprehension.
- The manuscript often cites reviews for primary concepts (e.g., the structure of the plant cell wall). While sometimes appropriate, it gives the impression of being a "review of reviews" rather than a critical synthesis of primary research. More direct citation of foundational and groundbreaking primary research is needed.
- There is repetitive language, such as "succinctly examined" followed by "reviewed" and "examined" in the abstract (lines 21-23). The writing could be more concise and dynamic.
- The transition to Section 4 feels tacked on. It should explicitly connect how nanomaterials can solve specific delivery challenges for CRISPR components (e.g., delivering large Cas proteins, protecting sgRNA).
- The "Bottleneck Landscape" Title is Unclear: Section 2's title is awkward. A clearer title would be "Limitations of Conventional Gene Delivery Systems."
- Line 48-56: The description of the plant cell wall as a barrier is good, but it should explicitly state the "size exclusion limit" (approx. 5-20 nm) here as a central design principle for all subsequent nanocarrier discussions.
- Line 91-92: The statement that monocots lack the biosynthetic pathway for signal molecules is an oversimplification. Some monocots do produce these molecules, and the problem is more complex (e.g., differential perception, production of inhibitors).
- Line 199-217: The opening of Section 3 would benefit from a table summarizing the different nanocarrier classes, their key properties (size, charge, cargo type), and demonstrated advantages/disadvantages.
- Line 288-290: The section on "Graphene Derivatives" lacks a strong hook. It should start by clearly differentiating graphene/GO from CNTs and why they are interesting (e.g., 2D structure, different interaction with biomolecules).
- Line 396-401: The finding that "particle internalization is not a prerequisite for achieving silencing" is fascinating and counter-intuitive. This deserves a more prominent discussion on the potential mechanisms (e.g., extracellular release, signaling) and its profound implications for nanocarrier design.
- Line 465-510: The section on peptide carriers is one of the more detailed, but it could be structured better by first introducing the major classes of CPPs used in plants and their general mechanisms before diving into specific examples.
- Line 520-554: The section on DNA nanostructures is forward-looking but ends with a list of challenges that applies to almost all nanocarriers. It should more clearly state what is unique about the challenges for DNA-based carriers (e.g., cost of production, stability in plant fluids).
- Line 639-677: The "Summary and Prospect" should be restructured. First, a concise summary of key findings. Second, a focused prospect on 3-4 top priorities (e.g., 1. Standardization of efficiency metrics, 2. Developing in planta targeting strategies, 3. Addressing regulatory and scaling hurdles).
Author Response

(The authors gave the same response as above.)

Round 2
Reviewer 3 Report
Comments and Suggestions for Authors
I accept this manuscript in present form as authors successfully answered my comments very well.